# TiO_2_–Based Nanofibrous Membranes for Environmental Protection

**DOI:** 10.3390/membranes12020236

**Published:** 2022-02-18

**Authors:** Cristina Ileana Covaliu-Mierlă, Ecaterina Matei, Oana Stoian, Leon Covaliu, Alexandra-Corina Constandache, Horia Iovu, Gigel Paraschiv

**Affiliations:** 1Department of Biotechnical Systems, Faculty of Biotechnical Systems Engineering, University Politehnica of Bucharest, 313 Splaiul Independentei, 060042 Bucharest, Romania; cristina_covaliu@yahoo.com (C.I.C.-M.); stoian_s_oana@yahoo.com (O.S.); leon.covaliu@yahoo.com (L.C.); constandache@yahoo.com (A.-C.C.); paraschiv2005@yahoo.com (G.P.); 2Advanced Polymer Materials Group, University Politehnica of Bucharest, 132 Calea Grivitei, 010737 Bucharest, Romania; horia.iovu@upb.ro

**Keywords:** polymers, membranes, electrospinning

## Abstract

Electrospinning is a unique technique that can be used to synthesize polymer and metal oxide nanofibers. In materials science, a very active field is represented by research on electrospun nanofibers. Fibrous membranes present fascinating features, such as a large surface area to volume ratio, excellent mechanical behavior, and a large surface area, which have many applications. Numerous techniques are available for the nanofiber’s synthesis, but electrospinning is presented as a simple process that allows one to obtain porous membranes containing smooth non-woven nanofibers. Titanium dioxide (TiO_2_) is the most widely used catalyst in photocatalytic degradation processes, it has advantages such as good photocatalytic activity, excellent chemical stability, low cost and non-toxicity. Thus, titanium dioxide (TiO_2_) is used in the synthesis of nanofibrous membranes that benefit experimental research by easy recyclability, excellent photocatalytic activity, high specific surface areas, and exhibiting stable hierarchical nanostructures. This article presents the synthesis of fiber membranes through the processes of electrospinning, coaxial electrospinning, electrospinning and electrospraying or electrospinning and precipitation. In addition to the synthesis of membranes, the recent progress of researchers emphasizing the efficiency of nanofiber photocatalytic membranes in removing pollutants from wastewater is also presented.

## 1. Introduction

Lately, oil spills as well as industrial wastewater discharges have led to major damage to water resources, seriously affecting human health and leading to an ecological imbalance. In this case, it is necessary to develop efficient technologies for the depollution of wastewater affected by oil spills and industrial wastewater discharges. Technologies such as filtration [1], adsorption [1,2], catalysis [3,4,5], centrifugation [6], electrocoalescence [7] or biological treatment [4] for wastewater treatment have been widely reported, studied, and applied.

In recent years, membrane-based technologies have been of interest and have been studied for wastewater treatment due to their relatively low cost, high separation efficiency and ease of operation. Membranes form barriers between two phases, so substances are transported selectively [8]. Depending on the structure of the membranes, they can be classified as porous membranes or dense membranes [9]. The selectivity of membranes and their transport properties are strongly dependent on the structure of their pores. The transport mechanisms are different depending on the porous or non-porous membranes and are presented in Figure 1 [10].

Nanotechnology has become one of the most important areas of research. Nanofibers with high size uniformity, large specific surface area, and high porosity are expected to become efficient nanomaterials for wastewater treatment. To prepare nanofibers, several techniques have been studied such as electrospinning [11], drawing [12,13], phase separation [14], template synthesis [15], self-assembly [16], interfacial polymerization [17], etc. Possibly the most attractive technology used to manufacture nanofibers is electrospinning due to its low cost and ease of operation.

After reviewing the synthesis methods, materials required and applications of electrospun-based fibers, the researchers proved that these fibers are potential materials for wastewater depollution applications. Due to the fact that titanium dioxide (TiO_2_) is the most widely known and studied photocatalyst, this article provides a guideline to the fabrication of electrospun-based TiO_2_ fibers, and their application in the degradation of organic pollutants. The article begins with electrospinning technology and a comparison with other composite fiber synthesis methods. This is followed by a comprehensive review of some recent electrospun-based TiO_2_ fiber applications in organic pollutants degradation and TiO_2_ modification.

In recent decades, more and more research has been done on electrospun nanofibrous photocatalytic membranes, also called ENPM. The photocatalyst has been integrated in the nanofibrous membrane, and this composite membrane has proven to be effective in degrading wastewater pollutants [18].

## 2. Electrospinning/Electrospraying Techniques

### 2.1. Electrospinning

Electrospinning is an electrohydrodynamic process by which nanofibrous materials are generated from the spinning of the polymeric dope solution using an electric field with a high potential [18]. The electrospinning system comprises four necessary components, namely a capillary or metallic needle, a syringe pump, a high voltage source and a ground conductive collector. The processing solution may contain a mixture of polymer powder (e.g., cellulose acetate, polyacrylonitrile), photocatalyst nanoparticles and solvent (such as N,N-dimethylformamide). The dope processing solution is introduced into the syringe pump. When a sufficiently high voltage is applied, the solution will come out in the form of fibers [19].

Electrospinning parameters that influence the average diameter, shape of the fiber, and its distribution can be represented by the properties of the solution (solvent properties, molecular weight polymer, or viscosity), jet formation parameters (flow rate, needle diameter, temperature, humidity, or electrical application) or the collection method (static, rotation, rotation speed, or disc collector) [20].

So far, over 200 natural and synthetic polymers, such as polyimide (PI) [21,22], polyacrylonitrile (PAN) [23,24], poly (vinylidenefluoride) (PVDF) [25], poly (vinyl alcohol) (PVA) [26], polylactic acid (PLA) [27], polyurethane (PU) [28], cellulose acetate (CA) [29], polyethylene oxide (PEO) [30], polycaprolactone (PCL) [31] etc., have been electrospun in nanofiber membranes [11]. Electrospun nanofiber membranes that were synthesized from a single polymer demonstrate poor separation capacity. Thus, to improve the membrane separation capacity, the raw materials used for the membrane synthesis and the surfaces of electrospun membranes can be modified. Changing the surface of the electrospun membranes by plasma treatment [32,33], low temperature [32], oxidation [24,34], or thermal crosslinking [35] demonstrates an increase in membrane performance. However, these improvement methods are limited in terms of durability and tolerance in practical applications. Thus, to effectively address the problem, the direct incorporation of some nanoparticles or functional polymers in the spinning precursor was performed [36,37,38,39]. Different types of materials are used for the electrospinning technique, but, depending on the application, they are selected considering their individual properties. The groups of materials that can be used in the electrospinning technique are presented in Figure 2 [18].

Coaxial electrospinning was introduced around 2003 to produce core-shell fibers, the properties of the material being different. Coaxial electrospinning technology is used to develop fibers composed of two types of polymers. The synthesis of core-shell fibers by the coaxial electrospinning technique has become more critical over time, given their potential in the field of wastewater treatment. In the future, this method may be more advanced to avoid any mixing of the polymers used as the core and shell. This technology ensures high mechanical properties, hydrophilicity, porosity, and prevents membrane swelling [40].

A versatile method employed to synthesize composite fibers is electrospinning. A power supply, a syringe pump, and a collector drum form the electrospinning apparatus. The electric potential which induces the formation of the Taylor jet cone and the ejection of the polymer solution from the Taylor cone is transmitted to the polymer solution in the syringe pump and the collector. The polymer fibers are produced by the elongation of the polymer solution followed by solvent evaporation. The fiber diameters are modified with the variation of the electrospinning parameters. Cossich et al. discovered that the formation of smooth fibers is due to the viscosity of the solution. The beaded fibers are formed when the polymer concentration is lower than 12 wt% of polyamide–12, while the smooth fibers were produced at higher concentrations of 12–18 wt% due to higher chain entanglement. In addition, the diameter of fibers increased in direct proportion to the concentration of the polymer [41]. This finding is confirmed by Mondal et al. who noted that as the concentration of poly(vinyl pyrrolidone) (PVP) increases, the diameter of the fibers also increases. The liquid jet tends to break and produce heavily beaded fibers when the polymer concentration is low [42].

Mondal et al. also showed that when the flow rate was increased, the diameter of the fibers also increases [42]. Soo et al. used response surface methodology (RSM) by varying the electrospinning parameters such as flow rate, applied voltage, and tip-to-collector distance (TCD) to optimize the diameter of the fibers. It was also observed that when an increasing voltage was applied, the diameter decreased linearly. Meanwhile, the diameter of the fibers showed a quadratic relationship with the polymer solution flow rate and TCD [43].

Soo et al. also showed that a larger Taylor cone is formed when the flow rate increases resulting in a thicker polymer jet, while thinner polymer fibers are produced when TCD distance is increased, due to a longer stretching time for the polymer jet to reach the collector [43]. Thus, the electrospinning of TiO_2_ nanofibers can be summarized as follows: (a) preparation of polymer template and TiO_2_ precursor, followed by (b) electrospinning of solution to form composite nanofibers, and lastly (c) calcination of electrospun fibers to remove the polymer template and to obtain crystalline TiO_2_ nanofibers [44].

### 2.2. Electrospraying

The electrospraying technique involves the use of electric forces to atomize the liquid material. Using a high electrical potential, the dope solution is thrown through the needle nozzle in the form of finely charged drops. The droplet size can be adjusted by adjusting the voltage and the feed rate parameters. Previous research indicates that the combination of the two techniques (electrospinning and electrospraying) leads to high efficiency in flexible photoelectrode manufacture for fuel cells [45] or solar cells [46].

Figure 3 presents a comparative schematic of electrospinning and electrospraying used for membrane manufacturing based on (±)-camphor-10-sulfonic acid (β) doped conductive polymer polyaniline (CSA/PANi) blended with polyethylene oxide (PEO) and poly (vinyl chloride) (PVC). To result in the CSA/PANi-PEO and PVC fibers, the distance between the spinneret and the collecting aluminum foil, the solution flow rate, the voltage, and the humidity were set. For the electrospinning experiment, a solution of TiO_2_ nanoparticles prepared in methanol at a certain concentration was also used. The CSA/PANi-PEO conductive membrane made of fibers was fixed on polyethylene and grounded [47].

The two techniques (electrospinning and electrospraying) can be combined in a single unit because they represent the same electrohydrodynamic process, this unit being named the simultaneous electrospinning and electrospray system (SEE) [48].

### 2.3. Techniques for TiO_2_ Electrospun Nanofiber Fabrication

In 2005, nanomaterials with hydrophilic surface properties were discovered by researchers. They studied the interactions that exist between the discovered nanomaterials and polymeric membranes. It was observed that the nanomaterials exhibit excellent electrostatic interaction, and when bonding to a polymeric material they favour hydrogen bonding. Thus, the attention of researchers has focused on nanomaterials-based modification methods [49,50,51]. The nanomaterials in the membrane can be captured by immersion methods or by mixing with casting solution. Researchers have shown that both methods significantly improve antifouling properties [52,53,54]. However, to optimize performance, proper nanomaterial selection as well as the loading percentage of nanomaterials in polymer membranes, is required [55,56,57].

#### 2.3.1. Electrospun TiO_2_ Nanofibers

Titanium dioxide (TiO_2_) can be embedded into electrospun nanofibers, and compared to other materials, it has advantages such as low cost, excellent photoactivity, and high chemical stability [58,59]. In photocatalytic applications, titanium dioxide has been the most studied semiconductor [60,61]. Most of the information provided by researchers is related to the production of titanium dioxide in powder or suspension form. The disadvantage is the difficulty to recycle titanium dioxide due to the difficult separation of the powder catalysts from the liquid, which results in a separation high cost [62]. To avoid this problem, inert supports can be functionalized using active semiconductor nanoparticles, thus eliminating the necessity of the post-filtration step, and permitting the reuse of the catalyst as long as its stability is maintained [63].

The synthesis of TiO_2_ and other supported catalysts includes several methods such as chemical/physical vapor deposition, sputtering, dip-coating, and sol–gel methods [64,65,66]. Moreover, several materials have been tested as supports such as glass, paper, ceramics, fiberglass, pumice stone, and stainless steel [67,68,69]. For manufacturing continuous nanofibers with uniform diameters, a wide range of compositions are used, and electrospinning is an easy and cost-effective method [60,70]. In fact, researchers are interested in one-dimensional (1D) nanostructured materials, such as TiO_2_ nanofibers, due to their high surface area, flexibility, and improved optical and photocatalytic properties [60]. Nanofibers containing TiO_2_ can be classified according to the technique used: (i) pristine TiO_2_ nanofibers, when the polymer solution is used to electrospin the TiO_2_; (ii) TiO_2_ coated nanofibers, when the electrospun nanofibers production is finished then the TiO_2_ is added; or (iii) modified TiO_2_ nanofibers, when another material is added to enhance the properties of the pristine TiO_2_ nanofibers [60].

Typically, the preparation of TiO_2_ nanofibers via electrospinning is accomplished by mixing a TiO_2_ precursor with a polymer, and to modify the TiO_2_ amorphous phase into a crystalline phase, the material is often calcined [71]. For instance, Nyangasi et al. produced nanofibers, which were calcined at 500 °C to generate TiO_2_ in anatase phase by mixing titanium isopropoxide with polymethylmethacrylate (PMMA) [72].

Nanofibers from TiO_2_ can be produced by electrospinning using polymer/semiconductor blended fibers as they are, even if it is quite common to use the TiO_2_ precursors and calcination. To produce pristine TiO_2_ nanofibers, strategies such as coaxial electrospinning and dual electrospinning are efficient [73]. As an example, to electrospin poly(vinyl alcohol) (PVA) and TiO_2_ nanoparticles simultaneously, Luo et al. [74] used coaxial electrospinning. Therefore, some instability and a de-mixing of both materials may occur in the nanofibers produced by a polymeric solution containing TiO_2_ (instead of a precursor) [75]. However, TiO_2_ nanofibers were successfully produced by Pasini et al. from a mixture of polyetherimide (PEI) and commercial TiO_2_ nanopowder in tetrahydrofuran (THF) and dimethylformamide (DMF). The produced fibers were submitted to cold plasma in the air and nitrogen atmospheres to improve their adhesion and photocatalytic performance. Thus, the photocatalytic activity of the material was judged to suitable through the discoloration of methylene blue which demonstrates a high stability, as the fibers maintained their ability even after five successive cycles [76].

One of the first approaches used to produce TiO_2_ nanofibers is to coat the pristine polymer electrospun nanofibers with TiO_2_ after the electrospinning process. Drew et al. prepared an aqueous solution containing titanium dioxide to immerse electrospun polyacrylonitrile fibers [77].

The TiO_2_ pristine nanofibers with different or enhanced properties are formed by the addition of transition metals and nonmetals, as in the TiO_2_ nanoparticle production [60]. For comparison, Kudhier et al. [78] manufactured pristine electrospun TiO_2_ nanofibers doped or not with silver. The authors observed a decrease in the material band gap and an increase in the antibacterial activity by adding Ag on TiO_2_ nanofibers. Similarly, by adding urea into the polymer solution, graphitic carbon nitride (g-C3N4)-doped TiO_2_ nanofibers were manufactured. An increase of the photocatalytic activity along with the charge recombination suppression was observed, by the formation of a heterojunction between the interfaces of TiO_2_ and g-C3N4 [79]. Polymer solution characteristics, work procedures, and ambient conditions strongly influence the nanofibers’ properties such as their diameter and morphology [60]. To produce nanofibers that grow in a unidirectional way, Kim et al. [80] used a modified aluminum collector. It was observed that the mechanical and optical properties of these unidirectional nanofibers improved, as they had a high crystallinity and efficient transport of charge-carriers. The influence of applied voltage, collector distance, solution flow rate, and polymer concentration were investigated by Someswararao et al. to produce thinner nanofibers. The nanofibers’ diameters were reduced, and their mechanical strength was also enhanced after the optimization of all parameters [70].

Moreover, the characterization of nanofibers is important to evaluate their structural, morphological, chemical, mechanical, optical, and thermal properties. The characterization of nanofibers is most commonly performed using the following techniques, namely X-ray diffraction (XRD), scanning electron microscopy (SEM) or transmission electron microscopy (TEM) with energy-dispersive spectroscopy (EDS), thermogravimetry (TGA), and differential scanning calorimetry (DSC), among others. All these properties are correlated with the nanofibers’ preparation methods and could enhance their environmental application capabilities. In addition, in order to not change the morphology and crystalline structures and consequently the functional properties of the electrospun TiO_2_ nanofibers that are subsequently calcined, it is important to adequately control the temperature. To produce TiO_2_ nanofibers used for photocatalytic purposes, temperatures of calcination suitable for the crystalline phase anatase will be used, these are lower than the ones involved in the formation of rutile or brookite phases, because anatase is the one that presents better photocatalytic properties and a weaker surface energy [70]. It is well known that calcination temperatures up to 500 °C lead to formation of the crystalline phase anatase, and between 600 °C and 800 °C the transformation to rutile phase occurs [70]. Pure rutile it is usually formed at calcination temperatures over 900 °C [60]. Kuchi et al. mixed titanium tetraisopropoxide (TTIP) and poly-vinyl pyrrolidone (PVP) and thus produced electrospun nanofibers at different calcination temperatures. Thus, at the heat treatment of 500 °C the crystalline phase formed was anatase, at a temperature of 700 °C a mix of anatase and rutile was formed, and at 900 °C only rutile was formed, this was confirmed by both XRD and Raman analysis. The applicability of the material is determined by the nanofiber’s properties, so they can serve various purposes such as biosensors, adsorption substrates, batteries, dye-sensitized solar cells, and catalysts for pollutants degradation [81].

Due to its high-photoactivity, good thermal and chemical stability, and low toxicity and cost, TiO_2_ is used more and more often in photocatalytic water treatment technology [82]. For pollutant degradation the immobilization of TiO_2_ in polymer has been investigated via many approaches, such as in polymer composite film, polymer gel, and polymer fibers. Due to their versatility for modification, high length to diameter ratio, and surface area, polymer fibers have been of great interest. The most utilized polymer for electrospun-based TiO_2_ fibers is poly(vinyl pyrrolidone) (PVP), while the supports in TiO_2_ have used various polymers and their combinations, this can provide advantageous characteristics such as high flexibility, stability in water, and spinnability. During photodegradation, fibers provide suitable support for TiO_2_ immobilization avoiding the loss of TiO_2_ and improving the contact between organic pollutants and TiO_2_. It also offers recyclability as another advantage, which can improve the economic value of the material. The recyclability of electrospun-based TiO_2_ fibers shows satisfying performance, whereby the performance drop can be minimized to around 6% after six times of photodegradation using the same sample [83], with some having their photocatalytic ability preserved completely after recycling [84]. Thus, to improve the performance of immobilized TiO_2_ and the effectiveness of surface contacts, the modifications of electrospun-based TiO_2_ fibers are still performed. This paper aims to discuss the continuous improvement of the modification of electrospun-based TiO_2_ fibers, covering the fabrication of electrospun-based fibers, the modification of TiO_2_, and the photoreactor utilized. Doping or loading of TiO_2_ with metal and non-metal elements or pairing TiO_2_ with other photocatalysts is recommended to improve its photodegradation performance. The recombination rate is reduced with these strategies. By transferring the electrons from the conduction band (CB) of TiO_2_ to the loading material or the other photocatalyst paired in the heterojunction system which will then react with the species adsorbed on surface of photocatalyst, the recombination rate will decrease the photocatalytic performance. Recently, several researchers have extensively reviewed synthesis methods, materials, and applications of electrospun-based fibers. For water treatment applications, the electrospun-based fibers proved to be potential materials. However, previous researchers have not clearly addressed the application of electrospun-based TiO_2_ fibers regarding the degradation of organic pollutants although TiO_2_, which is the most popular photocatalyst known [84].

#### 2.3.2. TiO_2_ Nanofibers

The researchers synthesized a p-n nanoheterojunction photocatalyst, consisting of zirconium nanofibers that were stabilized with yttrium (YSZ) to support silver oxide nanocrystals Ag_2_O and titanium dioxide nanoparticles TiO_2_, using a combination of electrophilation methods and precipitation. The process is shown in Figure 4 [85].

#### 2.3.3. Electrospun TiO_2_ Modification

In order to improve the efficiency of anatase titanium dioxide in the photodegradation process, doping with copper (II) sulfide and carbon can be performed [42,86,87]. Dai and Yin studied the carbon doping effect on anatase titanium dioxide in photocatalytic applications. They reported that the electron structure of anatase was not affected directly by carbon doping, but, under a slight stress, it denatured the “softened” anatase crystal’s lattice and induced the narrowing of the band gap, thus improving its optical capacity [86].

Mondal et al. argued that the purpose for which carbon acts as a dopant in titanium dioxide is to decrease its band gap due to surface states near the edge of the TiO_2_ valence band. When carbon concentrations are high, the photocatalytic activity decreases due to increased conductivity [42]. Zhang et al. [87] studied the use of copper (II) sulfide (CuS) for coating titanium dioxide fibers. Electrons from the valence band (VB) reach the conduction band (CB) of CuS, being excited due to fact that CuS absorbs the light. Due to the lower conduction band position of titanium dioxide, electrons are transferred to it and the holes will arrive from the titanium dioxide valence band into the CuS valence band. Since rutile has a smaller band gap in comparison with anatase, it can absorb more visible light. However, due to its smaller specific surface area, weak surface adsorption and larger grain size, rutile exhibits lower photocatalytic activity [88]. Lee et al. studied the use of titanium dioxide fibers, including graphene, for NOx photodegradation. Increased visible light degradation was reported using graphene due to the reduction of the band gap by creating oxygen vacancy in the titanium dioxide lattice. The location of possible carbon doping sites in titanium dioxide can be at the titanium site (cationic doping) or at the oxygen site (anionic doping) [88]. The graphene inclusion in the titanium dioxide lattice could produce a poorly packed Ti–O–Ti–C– polymeric skeleton, which impaired crystal growth, leading to oxygen depletion from the lattice and resulted in a lower band gap and increased visible light activity [88].

Photocatalytic activity can be improved by protecting electrospun fibers from photocatalytic degradation and the existence of an enhanced number of active sites, which is due to the realization of the multilayer structure of titanium dioxide fibers [89]. Lee et al. reported the fabrication of multilayer titanium dioxide fibers using plasma-treated poly(dimethylsiloxane-b-etherimide) (PSEI) to negatively charge the fibers resulting in acidic groups. After this process, alternating layers of polyhedral oligosilsesquioxane (POSS) which is positively charged, and TiO_2_ which is negatively charged, were added [89,90]. This method ensures the distribution of titanium dioxide on the surface of the fibers is uniformly achieved, ensures sufficient binding, improves the chemical or thermal properties of the fibers, creates a substance with a high specific surface area, and increases the resistance of the fibers to UV degradation [89,90]. In addition, another method was studied and used to enhance the binding of titanium dioxide nanoparticles on electrospun fibers using chelating agents as crosslinking agents. To chemically attach the nanoparticles of titanium dioxide to the surface of nanofibers of polyacrylonitrile (PAN), Chaúque et al. used ethylenediamine (EDA) and ethylenediaminetetraacetic acid (EDTA) as chelating agents. The accommodation of titanium dioxide nanoparticles attachment was achieved by reducing surface imines, amine, and carboxylic groups [91]. The attachment can be achieved by physico-chemical interactions, for example electrostatic attraction, hydrogen bonding, chelating and ester-like linkage, and bidentate bridging [91]. The improvement of methyl orange adsorption was achieved by the existence of combined electrostatic and non-electrostatic forces and thus, through photocatalytic degradation and adsorption the removal of methyl orange was improved [91]. Lian and Meng reported the fabrication of hybrid electrospun multilayer fibers made of TiO_2_/bioglass aiming to improve the interconnection of titanium dioxide fibers. These fibers were made of three layers, namely a bottom layer containing bioglass fibers, a middle layer containing TiO_2_-bioglass fibers and a top layer containing TiO_2_ fibers. The use of bioglass can improve the wettability and flexibility of electrospun fibers [92]. To fabricate a membrane, multiple layers of bioglass fibers and TiO_2_ fibers were spun using microfluidic channels to produce a multilayer fiber stack. This membrane was calcined at 500 °C to remove organic compounds and promote fiber fusion, maintaining a porous structure to achieve a “TiO_2_ wallpaper” [92]. The resulting porous structure showed an advantage in increasing the pollutants’ adsorption efficiency on the membrane surface. The TiO_2_ fibers present in the top layer of the “TiO_2_ wallpaper” photodegrade the adsorbed pollutants. Increasing the thickness would contribute with higher fiber stability and more available pores [92]. Lee et al. presented a simpler method of fabricating electrospun porous TiO_2_ fibers. The fibers were sonicated for 2 h after the electrospinning process. Using the bait-hook-and-destroy mechanism, which is a simultaneous photodegradation and adsorption process, the porous fibers were degraded methylene blue. A degradation efficiency of about 97% was reported after 240 min of contact time, and UV irradiation can completely remove methylene blue and all pollutants in less than 90 min [93]. A total of ten photodegradation cycles were performed to test the stability of the fibers, and after the last cycle the methylene blue was still completely removed. Pant et al. achieved a modification of the electrospun fiber structure using titanium dioxide by fabricating a spider web-like mat made of electrospun nanofibers. Due to the incorporation of titanium dioxide nanoparticles, the ionization of the polymer solution increased, inducing the formation of the spider web structure. It has been observed that the mechanical strength of these fibers is higher, the fibers show improved hydrophilicity and block imperative UV light, which are necessary for protective clothing applications [94]. The average photodegradation percentage of pollutants using electrospun-based TiO_2_ fibers is 86%, and the lowest degradation efficiency was 13% [95]. Most of the tests were conducted under visible light irradiation. Using a lamp as a light source provides a constant flux of photons as opposed to irradiating with sunlight [96]. Previous studies on the degradation of RB-5 and RhB under the irradiation of sunlight using TiO_2_/g-C_3_N_4_ with a sheet-like structure showed excellent results [97]. There are limited studies on the use of electrospun-based TiO_2_ applied in the process of photodegradation under irradiation with sunlight.

## 3. Applications of Electrospun Nanofibrous Membranes in Environmental Protection

The impressive performance of electrospun nanofibrous photocatalytic membranes for the degradation of pollutants from wastewater is shown in Table 1. ENPM can degrade harmful organic pollutants from wastewater. Due to the organic pollutant’s presence, the turbidity of the water could increase, and eutrophication could be achieved, reducing the amount of oxygen dissolved in the water. Even if the resulting yields are very good, in the development of electrospun nanofibrous photocatalytic membranes several challenges appear, namely high energy consumption, low flow rate, low mechanical stability or leaking of the photocatalyst from the substrate.

Surface functionalized electrospun nanofibers with active groups are responsible for the adsorption of toxic substances from wastewater or drinking water. Hota et al. reported the use of alumina/PVP nanofibers for the removal of chromium (VI) ions and fluoride. Two optimal pH values were reported, namely five and seven, the adsorption capacity being 6.8 and 1.2 mg/g during the 1 h of contact time [113].

Organic pollutants include herbicides, dyes, pharmaceuticals, humic substances, pesticides, phenolics, and petroleum, all of which pose a danger to the environment [114]. Organic pollutants appear in water due to anthropogenic activities, discharging water from agriculture or certain farms, residential water, or industrial wastewater [18].

### 3.1. Dyes Degradation

Wastewater with dye content is one of the main sources of pollution. Dyes are widely used in the food industry, in the manufacture of paper, paintings, textiles or cosmetics. Dyes are usually extracted from plants or insects and introduced into synthetic manufacturing processes [115].

To observe the photocatalytic activity of electrospun nanofibrous photocatalytic membranes, methylene blue is widely used in research as an organic dye. Vild. et al. reported that when using TiO_2_/PMMA electrospun fiber nanocomposites, methylene blue was almost 100% degraded from wastewater after a contact time of 100 min [20]. They applied Langmuir–Hinshelwood models as pseudo-first-order kinetic models to quantitatively determine the reaction kinetics.

Panthi et al. reported the use of 0.02 M silver phosphate immobilized nanoparticles (Ag_3_PO_4_) on electrospun PAN nanofibers for the degradation of methylene blue, and after a contact time of 60 min they observed the degradation of almost 100% of this organic dye [100]. The researchers observed that silver phosphate nanoparticles (Ag_3_PO_4_) attached to the nanofiber surface in agglomerated forms. Silver phosphate nanoparticles (Ag_3_PO_4_) are a promising photocatalyst for the photodegradation of organic pollutants. The mechanism of photodegradation of methylene blue by Ag_3_PO_4_/PAN composite nanofibers can be explained by the production of reactive oxygen species.

Due to the mutagenicity and toxicity effect of azo dyes, they have been banned worldwide. Azo dyes have vivid colors such as yellow, orange or red. Examples of azo dyes include Bismarck brown, aniline yellow, congo red or methyl orange [115]. Methyl orange (MO) is used as a model compound for the analysis of photodegradation of azo dyes [116,117]. Li et al. reported that the use of H_4_SiW_12_O_40_(SiW_12_)/cellulose acetate nanofiber membrane composite under UV irradiation for the photodegradation of methyl orange dye and the antibiotic tetracycline performed better than the use of neat SiW_12_ photocatalyst [98]. The optimum ratio of SiW_12_ to cellulose acetate was reported to be 1:4, and the efficiencies obtained for methyl orange degradation and tetracycline removal were 94.6% and 63.8%, respectively. It is observed that the percentage of degradation in the case of methyl orange was much higher (the difference between the degradation efficiency of methyl orange compared to that of tetracycline was more than 30%), under the same conditions. This may be due to the additional function of the cellulose acetate (CA) nanofibrous membrane that donated electrons to SiW_12_ during methyl orange degradation. Thus, this reaction increased the degradation efficiency of methyl orange in comparison to tetracycline. Moreover, the addition of the SiW_12_ catalyst to the electrospun CA substrate has improved its efficiency by providing a larger and more efficient contact area between the SiW_12_ photocatalyst and the pollutant.

Lian and Meng synthesized a “TiO_2_ wallpaper” composed of electrospun TiO_2_/bioglass hybrid nanofiber to degrade methylene blue from wastewater. The catalytic performance of the “TiO_2_ wallpaper” membrane was compared to that of the TiO_2_ film. The researchers also investigated the catalytic performance of the “TiO_2_ wallpaper” membrane by varying the thickness of its top layer. The “TiO_2_ wallpaper” membrane with a top layer of 0.051 mm demonstrated a degradation efficiency of methylene blue of about 60%, while the use of TiO_2_ film presented a degradation efficiency of only 33.5% after a contact time of 120 min. In this research it has been shown that the nanofibrous porous structure is still preserved when the “TiO_2_ wallpaper” thickness increases, thus ensuring the enlargement of the photocatalyst surface. Thus, the degradation of the pollutant could be even more efficient. On the other hand, the addition of several layers of TiO_2_ film was studied. Thus, up to four layers of film were added, but its porous structure could not be maintained, thus its photocatalytic performance was reduced [92].

### 3.2. Herbicides

Recently, to decompose organic pollutants, polyoxometalates (POMs) have been studied and used as photocatalysts [98,118,119]. Polyoxometalates (POMs) are represented by transition metal-oxygen groups and they are non-toxic, have controllable dimensions, have a high charge density, and have photostability. A disadvantage of POMs is that they are soluble in organic solvent and water, and their recovery and reuse are difficult. Furthermore, their specific surface area is small. Thus, polyoxometalate-compatible host materials could be established to give them the property of being recoverable and reusable and to improve their specific surface area for practical applications. Silicotungstic acid, H_4_SiW_12_O_40_(SiW_12_) is one of the polyoxometalates (POMs) that has been studied for the photocatalytic degradation of pollutants. Its Keggin structure (W-O-W) has contributed to its photocatalytic performance and good chemical stability [118].

Globally, the agricultural sector uses herbicides extensively to inhibit and control the growth of unwanted plants such as grass or weeds. The herbicides used for this purpose enter the soil, and are then released into groundwater. Thus, agricultural wastewater usually contains herbicides and will need to be treated before discharging or before being consumed by animals. In addition to dyes, phenylurea herbicides that have low toxicity (e.g., isoproturon) are efficiently degraded by ENPM [107]. The mechanism of isoproturon degradation using membranes composed of electrospun fibers—CQDs-Bi_20_TiO_32_/PAN is shown in Figure 5.

Isoproturon degradation was performed in three stages, namely: (1) the accumulation of isoproturon molecules on the fiber surface was encouraged by the membranes composed of electrospun fibers—CQDs-Bi_20_TiO_32_/PAN having a hierarchical mesoporous and macroporous structure, (2) in order to produce electrons ($e_{\mathrm{CB}}^{-}$) in the conduction band and photons ($h_{\mathrm{VB}}^{+}$) in the valence band, electrospun fiber membranes—CQDs-Bi_20_TiO_32_/PAN, were used as photocatalysts to adsorb the energy of visible light on the surface of the fiber. The electrons in the conduction band ($e_{\mathrm{CB}}^{-})$ migrate to the CQDs and generate radical O^•^_2_^−^ following the reaction of the electrons with oxygen, and (3) isoproturon molecules are completely degraded by the active species O^•^_2_^−^ and photons in the balance band ($h_{\mathrm{VB}}^{+}$) and harmless CO_2_ and H_2_O are formed.

### 3.3. Polymers

The discharge of untreated polymers in the form of fluid injected into water bodies lead to damage to the environment, aquatic life, and human health. Degradation of polymers in the aqueous system and in the environment is very slow due to their complex structure. By creating more intermediate products, the photodegradation of polymers could be more complex before their degradation into harmless CO_2_ and H_2_O. Hashim et al. synthesized electrospun GCN/PAN nanofibers by sonicate, a solution made of DMF and GCN powder, for 5 h to which PAN powder was subsequently added. The final solution was sonicated for another 5 h. The resulting solution was subjected to the electrospinning process to obtain GCN/PAN nanofibers, as shown in Figure 6. A stainless-steel plate was used as a collector for the electrospun nanofibers which was wrapped in aluminum foil [108].

Hashimah et al. presented the intermediate products generated by the partially hydrolyzed polyacrylonitrile (HPAM) photodegradation using GCN/PAN nanofibers. The process is shown in Figure 7 [108]. To degrade partially hydrolyzed polyacrylonitrile (HPAM) into intermediate (propionamide), GCN generated hydroxyl radicals were placed under UV irradiation, then the alkyl group was removed to provide another intermediate product (acetamide). The alkyl group reacts with H^+^ in another segment of HPAM to produce acetic acid. In addition, during the degradation of HPAM using GCN/PAN nanofibers, ammonia was formed and transformed by photocatalysis into a nitrite ion and then degraded to nitrate. After 180 min of contact time under UV light irradiation, GCN/PAN nanofibers showed a degradation efficiency of HPAM of 90.2%.

### 3.4. Pharmaceuticals Removal from Wastewater

Qayum et al. used the fibrous membrane PAN/AgBr/Ag and studied its photocatalytic activity to degrade salicylic acid from wastewater with visible light [106]. The membrane was manufactured in 5 h and demonstrated a salicylic acid degradation efficiency of approximately 97%. The researchers also reported that even after the fifth cycle of use, the PAN/AgBr/Ag membrane demonstrated a salicylic acid degradation efficiency of up to 96%.

Usually, before organic pollutants are completely removed from pharmaceutical, antibiotic or oilfield wastewater, the process of photodegrading generates intermediate by-products that are harmful. The reaction mechanism in the photodegradation of organic pollutants reflects the generation of these intermediate by-products. An example is phenol, which contains a benzene ring structure that is very stable and resistant to decomposition. Two main phases are involved in the photodegradation of phenol using TiO_2_, namely the intermediate and mineralization phases [120]. In the first (intermediate) phase several intermediate compounds can be formed by the transformation of phenol, such as resorinone, catechol or benzoquinone, following the addition of the hydroxyl group in the phenol structure in the para or ortho position. Thus, the ring structure of phenol is open which leads to the conception of hydrocarbon chains. Reactive oxygen species (^•^OH, $O_{2}^{-}$, H_2_O_2_ and ^1^O_2_) easily oxidize short and weak hydrocarbon chains and form CO_2_ and H_2_O, thus completing the phenol mineralization phase.

The presence of paracetamol in the environment, as a pharmaceutical compound, has attracted the attention of researchers as being a potential contaminant for water. Also called 4-hydroxyacetanilide, paracetamol is usually used as an antipyretic and analgesic agent to reduce fever and pain. However, liver failure and even death can be caused if paracetamol is used excessively. The researchers reported that it is difficult to achieve complete degradation of paracetamol due to the existence of a stable aromatic ring in its structure [121,122]. Moctezuma et al. reported paracetamol photodegradation using TiO_2_ under UV irradiation. The process was monitored by UV-vis, total organic carbon (TOC), fourier transform infrared spectroscopy (FTIR), and high-performance liquid chromatography (HPLC) analysis [122]. Thus, the formation of intermediate aromatic by-products h been discovered, such as p-nitrophenol and p-aminophenol which eventually mineralizes. It has been observed that through an alternative diacylation mechanism by which slightly oxidized p-aminophenol develops into p-nitrophenol, converting to hydroquinone and nitrocathecol, the oxidation of paracetamol can occur. Before being completely degraded to CO_2_ and H_2_O, benzoquinone and hydroquinone are oxidized to carboxylic acids with a low molecular weight.

Potential secondary pollution may occur due to incomplete or partial degradation of organic pollutants. Thus, before discharging into water bodies, it is necessary to completely photodegrade organic pollutants using photocatalysts. The organic pollutants’ complete degradation can be achieved using electrospun nanofibrous photocatalytic membranes (ENPM). They support the photocatalyst to achieve the adsorption of pollutants from wastewater on the surface of the photocatalyst before the photodegradation process. However, in the case of overloading the photocatalyst in the ENPM membrane support, leakage of pollutant which has been adsorbed by the photocatalyst and which shows incomplete degradation may occur, being a secondary pollutant. Thus, it is necessary to optimize the performance of ENPM membranes without damaging their physico-chemical properties, by optimizing the photocatalyst load on these membranes.

## 4. Conclusions

The electrospinning technique is an advantageous method for the manufacture of nanofiber membranes generated from different polymers. The morphology of the resulting membranes is determined directly by the parameters of the solution and the operating conditions. Nanofibers produced by the electrospinning technique are an attractive choice for environmental applications. Through the electrospinning technique it is possible to produce membranes with relatively uniform pore size distribution and high pore interconnectivity, compared to conventional techniques. To increase the functionality of membranes obtained by electrospinning, research has led to the incorporation of functionalizing agents (i.e., nanoparticles) in these membranes which has improved their performance for certain applications such as removing bacteria, viruses, or metal ions from wastewater.

Membranes obtained by electrospinning can be post-treated, chemically or thermally, with the aim of modifying characteristics such as hydrophobicity, pore size, mechanical integrity, or electrical conductivity.

Fabricating TiO_2_ fibers and thus immobilizing TiO_2_ has been highly applied in the photodegradation of wastewater pollutants. In order to modify the morphology of the fibers, improve their properties and fabricate tough composites, electrospinning is combined with other methods (electrospray, coaxial electrospinning).

For the degradation of pollutants, the most widely used photocatalyst is TiO_2_. It shows good performance in dye degradation and has good photochemical stability in comparison to other types of photocatalysts. The anatase/rutile performance was observed to be greater than that of anatase on its own for photodegradation, therefore, calcination has a role in inducing phase transformation. Carbonization or calcination of electrospun TiO_2_ may improve crystal growth, create fiber porosity, and modify surface roughness, which affects the fibers’ hydrophilicity.

ENPM membranes made of various polymeric materials are used in water treatment due to their efficiency, accessibility, durability, higher surface area-volume ratio, higher reactivity, operation without the addition of chemicals, and smaller pore size.

The impressive capacity of electrospun nanofibrous photocatalytic membranes (ENPM) for degradation of harmful organic pollutants is shown in Table 1. It is observed that the use of an ENPM to remove methylene blue from wastewater showed a 100% treatment efficiency in only 60 min contact time. In addition, by using various ENPMs, a complete degradation of wastewater from Propranolol, Cimetidine (CMT), 4-Chlorophenol, and Bisphenol A was achieved in 60, 40, 100 and 100 min, respectively. However, the development of ENPM has presented several challenges such as neglected filtration properties, low throughput, high energy consumption, low mechanical stability, or leakage of the photocatalyst from the substrate.

According to this review, most research has shown that nanofiber photocatalytic membranes should be introduced into wastewater for the photocatalytic degradation of pollutants, but their use as filter membranes has been neglected.

Even if huge progress has been made, further research can be done to achieve better performance of electrospun membranes depending on the solution parameters and operating conditions.

## 5. Challenges

Based on this study, most of the research articles showed that the manufactured electrospun nanofibrous photocatalytic membranes were immersed directly into wastewater for photocatalytic degradation of pollutants, while filtration, the main function of the membrane, was neglected. This restricts the photocatalytic degradation efficiency of the manufactured nanofibrous photocatalytic membrane. There are quite a few articles showing the contribution and significance of filtration on pollutants photocatalytic degradation by nanofibrous photocatalytic membranes. It is still unclear why filtration improves the photocatalytic degradation efficiency. This is a scientific question that will need to be addressed by further research. Thus, systematic studies will need to be carried out to address the practical applicability of nanofibrous photocatalytic membrane materials.

## Figures and Tables

**Figure 1 membranes-12-00236-f001:**
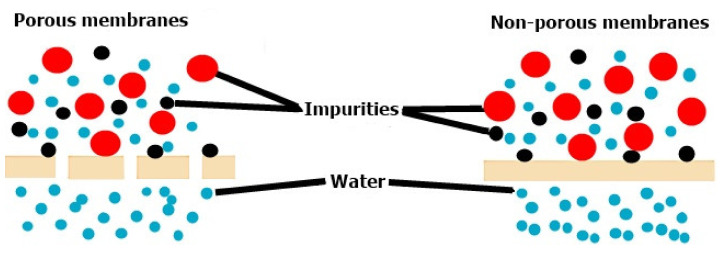
Transport mechanisms using a porous membrane (**left**) and a non-porous membrane (**right**) [10]. Reproduced with permission from Farah Ejaz Ahmed, Boor Singh Lalia, Raed Hashaikeh, A review on electrospinning for membrane fabrication: Challenges and applications; published by Elsevier, 2015.

**Figure 2 membranes-12-00236-f002:**
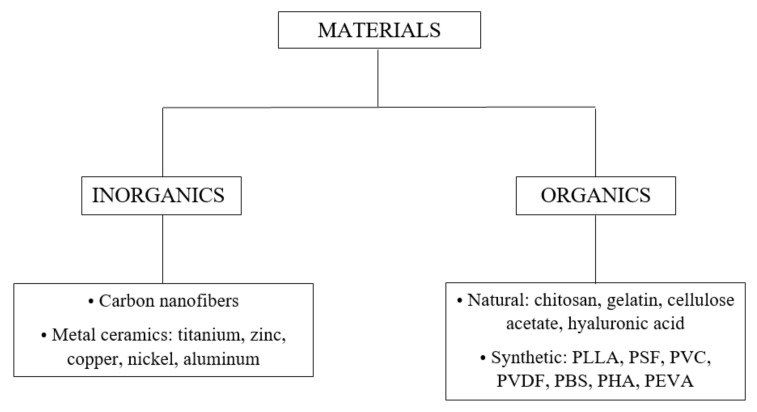
Classes of materials generally used for the electrospinning technique. Abbreviations: PLLA—polylacticco-glycolic acid, PSF—polysulfone, PVC—polyvinyl chloride, PVDF—polyvinylidene fluoride, PBS—polybutylene succinate, PHA—polyhidroxyalkanoates, PEVA—polyethylene-co-vinyl alcohol.

**Figure 3 membranes-12-00236-f003:**
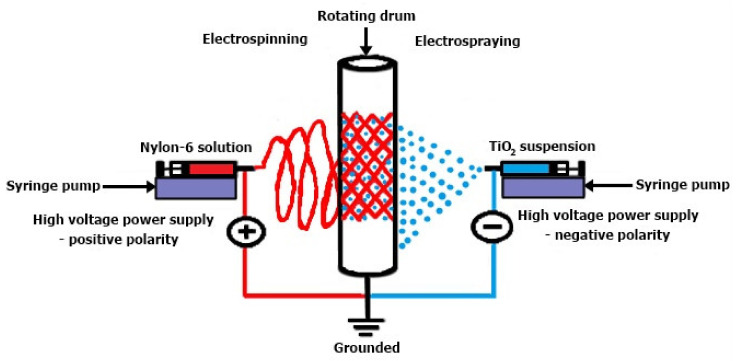
The SEE—simultaneous electrospinning and electrospray system diagram [48]. Reproduced with permission from Atikah Mohd Nasir, Nuha Awang, Juhana Jaafar, Ahmad Fauzi, Ismail, Mohd Hafiz Dzarfan Othman, Mukhlis A. Rahman, Farhana, Aziz, Muhamad Azizi Mat Yajid, Recent progress on fabrication and application of electrospun nanofibrous photocatalytic membranes for wastewater treatment: A review; published by Elsevier, 2021.

**Figure 4 membranes-12-00236-f004:**
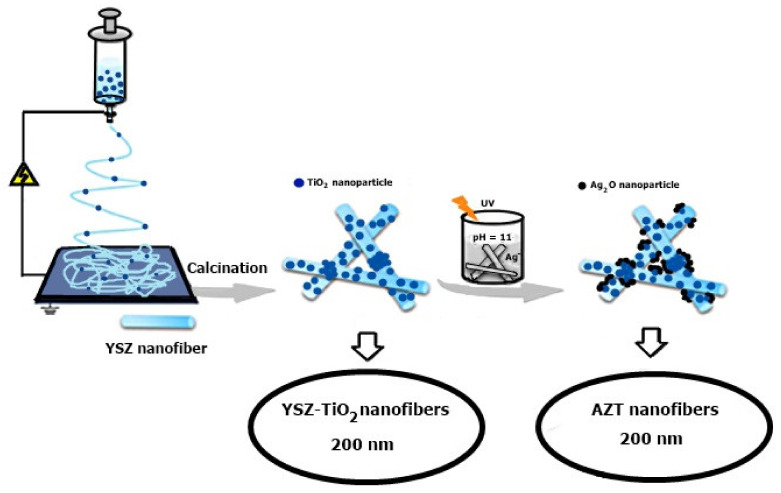
Manufacture of Ag_2_O@YSZ-TiO_2_ by combining electrospinning and precipitation methods [85]. Reproduced with permission from Atikah Mohd Nasir, Nuha Awang, Juhana Jaafar, Ahmad Fauzi, Ismail, Mohd Hafiz Dzarfan Othman, Mukhlis A. Rahman, Farhana, Aziz, Muhamad Azizi Mat Yajid, Recent progress on fabrication and application of electrospun nanofibrous photocatalytic membranes for wastewater treatment: A review; published by Elsevier, 2021.

**Figure 5 membranes-12-00236-f005:**
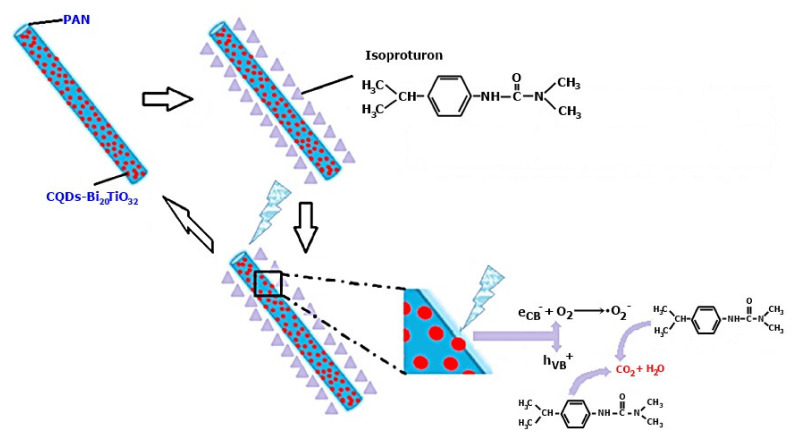
Isoproturon degradation using electrospun fiber membranes—CQDs-Bi_20_TiO_32_/PAN [107]. Reproduced with permission from Atikah Mohd Nasir, Nuha Awang, Juhana Jaafar, Ahmad Fauzi, Ismail, Mohd Hafiz Dzarfan Othman, Mukhlis A. Rahman, Farhana, Aziz, Muhamad Azizi Mat Yajid, Recent progress on fabrication and application of electrospun, nanofibrous photocatalytic membranes for wastewater treatment: A review; published by Elsevier, 2021.

**Figure 6 membranes-12-00236-f006:**
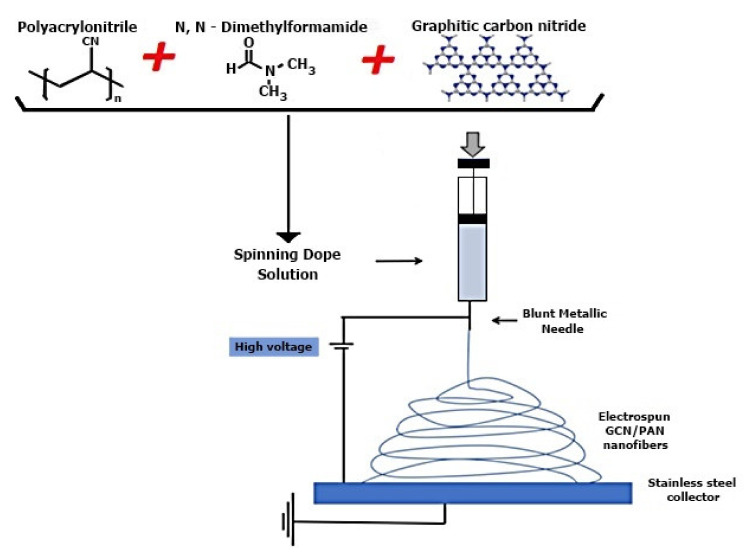
Formation of electrospun nanofibers—GCN/PAN by the electrospinning technique [108]. Reproduced with permission from Nur Hashimah Alias, Juhana Jaafar, Sadaki Samitsu, A.F. Ismail, M.H.D. Othman, Mukhlis A. Rahman, Nur HidayatiOthman, N. Yusof, F. Aziz, T.A.T. Mohd, Efficient removal of partially hydrolysed polyacrylamide inpolymer-flooding produced water using photocatalytic graphiticcarbon nitride nanofibres; published by Arabian Journal of Chemistry, 2020.

**Figure 7 membranes-12-00236-f007:**
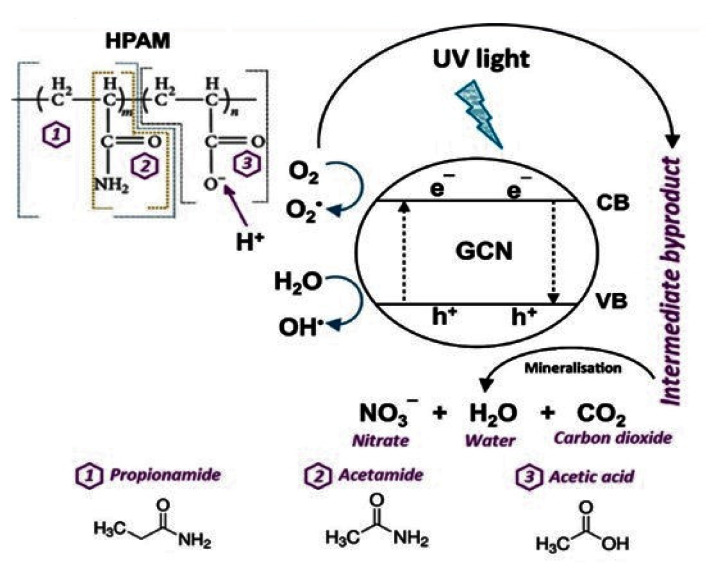
Generation of intermediate by-products from the photodegradation process of HPAM using GCN /PAN nanofibers [108]. Reproduced with permission from Atikah Mohd Nasir, Nuha Awang, Juhana Jaafar, Ahmad Fauzi, Ismail, Mohd Hafiz Dzarfan Othman, Mukhlis A. Rahman, Farhana, Aziz, Muhamad Azizi Mat Yajid, Recent progress on fabrication and application of electrospun, nanofibrous photocatalytic membranes for wastewater treatment: A review; published by Elsevier, 2021.

**Table 1 membranes-12-00236-t001:** Recent research on fabrication of ENPM and in their application for the degradation of pollutants from wastewater.

ENPM	Synthesis Method	Pollutant	ENPM Quantity (g)	Pollutant Concentration (mg/L)	Solution Volume (mL)	Degradation Time (min)	Photocatalytic Degradation (%)	Ref.
H_4_SiW_12_O_14_/CA	Electrospinning	Methyl orange	0.20	10	100	120	94.60%	[98]
TiO_2_/bioglass nanofibers	Electrospinning, calcination, heating	Methylene blue	-	10	-	120	60.0%	[92]
TiO_2_/PMMA	Electrospinning	Methylene blue	-	2	50	100	100%	[20]
PAN/TiO_2_/Ag	Electrospinning and hydrothermal	Methylene blue	0.01	10	20	60	99.7%	[99]
Ag_3_PO_4_/PAN nanofibers	Electrospinning and surface modification	Methylene blue	0.15	10	50	60	100%	[100]
Pdopa-ZNRs/PU	Electrospinning, surface functionalization and hydrothermal	Methylene blue	-	10	20	180	65.0%	[101]
CNF@TiO_2_	Blended spinning and carbonization	Rhodamine B	-	10	200	60	80.0%	[102]
MOF-based C-doped coupled TiO_2_/ZnO nanofibers	Electrospinning and calcination	Rhodamine	0.02	10	25	100	92%	[103]
TiO_2_/g-C3N4 heterojunction	Electrospinning and calcination	RhB	0.05	10	50	100	96.0%	[104]
rGO@TiO_2_	Electrospinning and calcination	Propranolol	-	5	50	60	100%	[105]
H_4_SiW_12_O_14_/CA	Electrospinning	Tetracycline	0.20	10	100	120	63.80%	[98]
PAN/AgBr/Ag	Electrospinning, heat treatment and wet chemical	Salicylic acid (SA)	0.10	5	20	300	97.0%	[106]
CQDs-Bi_20_TiO_32_/PAN	Coaxial electrospinning	Isoproturon	0.10	15	50	3600	90.4%	[107]
Graphitic carbon nitride/PAN	Electrospinning	HPAM	-	20	-	180	90.2%	[108]
PVDF-TiO_2_	Electrospinning and electrospraying	Cimetidine (CMT)	-	-	45	40	100%	[109]
		4-Chlorophenol	-	-		100	100%	
		Bisphenol A	-	-		100	100%	
GO/ZnS-CNFs	Electrospinning, calcination and solvothermal	p-aminotoluene	-	-	10	n/a	90.0%	[110]
Graphitic carbon nitride Nanofibers	Electrospinning	Oilfield produced water	0.20	1000	200	480	96.6%	[111]
GCN/PAN nanofibers							85.4%	
Nanofiber coated alumina	Electrospinning and coating	Oilfield produced water	-	1000	-	180	99.0%	[112]

## Data Availability

Data sharing is not applicable.

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
