# Peer review of "TiO2–Based Nanofibrous Membranes for Environmental Protection"

_membranes, 2022, doi:10.3390/membranes12020236_

Round 1

Reviewer 1 Report

The authors have addressed my main concern with their manuscrip and have re-contextualized their review within the application of TiO2-based electrospun nanofibers. I believe that the paper is now suitable for publication.

Author Response

Esteemed reviewer,
Thank you for your response and acceptance.

Reviewer 2 Report

Hello,

In general, the article is tackling a very important and interesting topic, as it discusses the application of TiO2-based nanofibrous membranes for environmental protection. However, some comments as summarized in the attached report have to be considered to make it suitable for dissemination.

Regards

Author Response

Esteemed reviewer,
In the attached file you will find the revised article. Also, in the following lines we have responded to the points highlighted by you.
Thank you!

Review comments for membranes- 1584879-v1

Nanofibrous membranes based on TiO2 for environmental protection

In general, the article is tackling a very important and interesting topic, as it discusses the application of TiO2-based nanofibrous membranes for environmental protection. However, some comments summarized as follow have to be considered:

  • I would suggest modifying the title to “TiO2-based Nanofibrous…..”, rather than “Nanofibrous membranes based on …..”

Answer: The change was made to row 2.

  • The abstract has to include some information in relation to TiO2-based nanofibrous membranes, and their specific importance hence selected to produce this article to discuss it. Additionally, the abstract could include some unique performance characteristics of such membranes.

Answer: Additions have been made and can be seen on lines 17-21.

  • I would suggest having an introduction section that generally introduces the article rather than discussing some core topics as presented in subsections 1.1, 1.2…. Instead, these sections (i.e. 1.1, 1.2…) can be split into a new section on electrospinning/electrospraying techniques.

Answer: Additions have been made and can be seen on lines 51-63 and the new section “electrospinning/electrospraying techniques” was introduced at line 68.

  • The abstract and introduction section has to clearly present the novelty or contribution of this article, which is not currently present.

Answer: The contribution to this article has been highlighted in the abstract on lines 21-25 and in the introduction on lines 56-59.

  • “TiO2 as photocatalyst integrated into membranes”, has nothing to do with the manuscript, additionally, it has not been discussed. I would recommend such subtitles in this section.

Answer: The change has been made.

  • Figures should be of higher quality or redrawn as necessary.

Answer: The change has been made.

  • In line #135, it should be “schematic”, not “example”.

Answer: The change has been made and can be seen on line 145.

  • Some proofreading is required.

Answer: Changes have been made.

  • Applications of electrospun nanofibrous membranes in environmental protection:
  1. The applications should be critically and thoroughly discussed.
  2. Applications should be subdivided into different sections according to application class or family.
  3. Furthermore, application to protect against organic pollutants can be subdivided according to the family of organics such as dyes, aromatics, PAHs, BTEX…. etc.
  4. In table 1, a complete set of removal conditions should be provided.

Answer: The discussion of applications can be found on lines 404 - 546, specifically from page 10 to page 14. The applications have been subdivided and can be seen in rows 399, 449, 481, 503. The table has also been completed.

  • A section should be added to discuss the challenges facing the application of nanofibrous membranes for environmental protection, as well as the way forward or research needs.

Answer: The challenges section was added last time but we notice that it is no longer in the existing manuscript in the platform and therefore we have added this section again and it can be seen in lines 601-611.

  • More constructive conclusions should be provided

Answer: Several conclusions were added last time but we note that they are no longer in the existing manuscript in the platform and, therefore, we have added these conclusions again and they can be seen in lines 572-582.

Round 2

Reviewer 2 Report

Hello,

Thanks to the authors for considering the given comments.

Regards